# The Effects of Olfactory Loss and Parosmia on Food and Cooking Habits, Sensory Awareness, and Quality of Life—A Possible Avenue for Regaining Enjoyment of Food

**DOI:** 10.3390/foods11121686

**Published:** 2022-06-08

**Authors:** Alexander Wieck Fjaeldstad, Barry Smith

**Affiliations:** 1Department of Otorhinolaryngology, University Clinic for Flavour, Balance and Sleep, Regional Hospital Gødstrup, Hospitalsparken 15, 7400 Herning, Denmark; 2Department of Clinical Medicine, Flavour Institute, Aarhus University, Palle Juul-Jensens Boulevard 99, 8200 Aarhus, Denmark; 3Center for Eudaimonia and Human Flourishing, Linacre College, University of Oxford, Oxford OX3 9BX, UK; 4Centre for Olfactory Research and Applications, Institute of Philosophy, School of Advanced Study, University of London, London WC1E 7HU, UK; barry.smith@sas.ac.uk

**Keywords:** olfactory loss, parosmia, food liking, cooking

## Abstract

Olfactory dysfunction often has severe consequences on patients’ quality of life. The most common complaint in these patients is their reduced enjoyment of food in both patients with olfactory loss and parosmia. How the different types of olfactory dysfunction differ in relation to food and cooking habits, sensory awareness, and food-related quality of life has not yet received much attention. By applying questionnaires on cooking, food, olfactory function, weight changes, sensory awareness, and food-related quality of life, we investigated how various aspects of eating differ between participants with olfactory loss (*n* = 271), parosmia (*n* = 251), and normosmic controls (*n* = 166). Cooking habits in olfactory dysfunction revealed pronounced differences as compared with normosmic controls. Cooking with olfactory dysfunction was associated with, e.g., a lack of comfort and inspiration for cooking and an inability to make new foods successfully. Significant differences in cooking were also found between olfactory loss and parosmia. Food items were less familiar in participants with olfactory loss and parosmia, while the ratings of liking food items differed between olfactory loss and parosmia, indicating the importance of adapting ingredients in meals separately for olfactory loss and parosmia. Parosmia was associated with a higher incidence of weight loss, but we found no difference in food-related quality of life between participants with olfactory loss and parosmia. While olfactory loss and parosmia have wide-ranging consequences on patients’ cooking and food habits, adapting meals to include ‘safer food items’ and integrating multisensory stimulation may be a possible avenue for improving the enjoyment of food.

## 1. Introduction

Olfactory disorders are quite common in the general population, where 15% suffer from a reduced sense of smell while around 2% suffer from complete loss of smell [1]. In the wake of the COVID-19 pandemic, awareness of olfactory disorders has reached unprecedented levels [2] with around 65% of over 300 million COVID-patients world-wide experiencing sudden smell loss, often with long-lasting effects on both olfactory sensitivity and olfactory distortions [3,4].

While loss or reduction of smell can have detrimental effects on several aspects of daily life ranging from social insecurity, increased risk of depressive symptoms to increased risk of household accidents [5], the predominant quality of life complaints among patients are effects of the decreased enjoyment of food and other food-related problems [6]. 

Olfaction plays a key role in our ability to enjoy food, as olfactory cues and odour perception contribute significantly to flavour perception, food preference, acceptability, and intake. Food odours released in the oral cavity travel to olfactory receptors in the nose. In this way, retronasal olfaction makes an essential contribution to multisensory perception and identification [7]. Patients with a reduced sense of smell have been shown to be less attracted to novel foods and experience less pleasure when eating [8]. The consequences of olfactory loss on food extend beyond a lack of pleasure. At a population-level, there is an association between poor diet quality and lower variation in dietary habits in patients with olfactory dysfunction [9]. 

Olfactory dysfunction can be subcategorized quantitatively in olfactory loss, where the sense of smell can be reduced (hyposmia) or absent (anosmia), and qualitatively in olfactory distortions, where the perception of odours is distorted (parosmia) or odour experiences occur without an odour source (phantosmia). While most studies on olfactory dysfunction and food have been focusing on the quantitative loss of smell, recent advances have been made in understanding the effects of olfactory distortions. In these patients, specific patterns of distortions have been identified based on the molecular composition of odours [10]. Here, the particular focus has been on identifying the negatively perceived odours that trigger olfactory distortions. These distortions can have grave consequences on the desire and ability to eat and prepare food, appetite loss, weight change, and pleasure taken in food [11].

The tasting of flavours is a multisensory process, where each sense contributes notes in the composition of a complex symphony. Different senses contribute to the overall perception that enables us to identify what we eat, evaluate its freshness and edibility, and give rise to pleasure. A potato chip without its crunchy sound can ruin the experience and desire to eat [12] just as the lack of aroma can spoil the enjoyment of food in patients with olfactory dysfunction. However hard to imagine, the familiar sound of a violin could perhaps be replaced with a cello without entirely ruining the overall experience of the complex symphony. In a similar way, could the overall experience and pleasure of a meal for patients with olfactory dysfunction be partly mitigated by an increased compensatory focus on other sensory inputs? In order to achieve this aim, a solid understanding of the food-related consequences of olfactory dysfunction is a prerequisite along with an understanding of how other senses can augment the disrupted sensory experience of eating. 

Our aim in the current study is to investigate the consequences on food-related quality of life in patients with olfactory dysfunction and to what extent this is affected by the presence of parosmia. Furthermore, we aim to explore how these two types of olfactory disorders differ from a normative population in regard to cooking habits, food recognition, intake, and liking, and focus on other sensory properties of food. In comparison with recent studies, we do not aim to identify specific parosmic triggers or negative olfactory experiences, but instead, identify food items with the least decrease in liking that can be used in cooking an enjoyable meal.

## 2. Materials and Methods

### 2.1. Recruitment and Ethics

An online questionnaire was designed in REDCap (version 11.1.29) [13] and distributed online on social media and on flyers in waiting rooms of general practitioners and hospital outpatient clinics. Participants were eligible to fill out the survey if they were 18 years of age or above and had either a subjectively assessed normal sense of smell (controls) or were suffering from olfactory dysfunction. 

The study was conducted according to the Declaration of Helsinki on Biomedical Research Involving Human Subjects. All respondents have consented to participation in the study. No personal-identifiable data on any participant was obtained in the questionnaire (e.g., name, birth date, email, IP address, or social security number). The questionnaire-based design of the study did not require ethics approval according to Danish law (Danish Committee law §14.2), which was confirmed by the Regional Ethics Committee.

### 2.2. Data Collection

#### 2.2.1. Part 1—Demographics and Sensory Function

The questionnaire included baseline information on demographics (country of residence, age, sex), sense of smell (normal or olfactory disorder), degree of subjective quantitative olfactory loss (0–100, VAS scale), occurrence and severity of parosmia and phantosmia (none, rarely, often, always), aetiology of olfactory loss, subjective quantitative gustatory function (0–100, VAS scale), occurrence and severity of parageusia and phantogeusia (none, rarely, often, always), duration of olfactory disorder (months), weight change due to olfactory dysfunction (none, increase, decrease, more fluctuations; kg), the importance of food to quality of life (current and before olfactory disorder (0–100, VAS scale)), severity of olfactory disorder-related effect on quality of life (0–100, VAS scale), changed awareness of other senses following olfactory disorder (5-point Likert scale). 

#### 2.2.2. Part 2—Liking, Recognition, and Frequency of Intake of Basic Tastants and Food Items

On a 5-point Likert scale, participants rated the recognition, liking and frequency of intake for spicy food, umami, different types of sweet, sour, salty, umami, trigeminal/spicy, fatty and aromatic ingredients, and ratings of common pleasure-yielding foods/beverages and known parosmia triggers. Recognition was rated by asking the participant the following question: ‘If you were served the food item, would you be able to recognise what it was using taste, smell, and mouthfeel combined?’. Liking was rated for the current liking of food items ranging from disgust to pleasure.

In the multisensory domain of flavour perception, several senses may play a collective role in compensating for olfactory deficits. However, we chose to focus on ingredients with key basic tastant, mouthfeel, or aromatic attributes.

The list of items was created in collaboration between olfactory scientists and chefs working with a cooking school for patients with smell loss with the intention of identifying both ‘safe’ and ‘unexplored’ foods/ingredients, which could have potential for application in meals designed for patients with olfactory dysfunction. 

#### 2.2.3. Part 3—Cooking and Food Habits

The validated Cooking and Food Provisioning Action Scale [14] questionnaire for the measurement of individual cooking practice was administered to all participants. The 28-item questionnaire quantifies the relative capacity to plan and prepare meals. 

The CAFPAS includes three subscales that measure different aspects of the participant’s self-conceived ability to plan and achieve meal-related goals. 

The CAFPAS includes three different subscales, The 13-item “Skill and Self-Efficacy” subscale (the abilities and skills surrounding cooking adequate), 10-item “Attitude” subscale (the affective stance towards food, cooking, and provisioning), and the 5-item “Structure factors” subscale (if external, structural factors hinder or support cooking and provisioning actions and goals). Of the 28 items, 11 items are negative statements, and are reversed in the calculation of the subscale sum. To calculate a subscale score, the subscale sum is divided by the standard deviation of the whole sample population’s scores on the subscale. The total CAFPAS score is the calculated sum of all three subscales [14].

### 2.3. Data Analysis

Data on demographics was calculated as a proportion (%) and count (*n*). Differences in olfactory dysfunction-related quality of life between the parosmia group and olfactory loss group were calculated using a two-tailed Student’s *t*-test. MANOVA was used to calculate differences between food-related current quality of life, CAFPAS score differences, and food item liking, recognition and intake between the two olfactory dysfunction groups and the normosmic group. Principal component analysis was used to calculate correlations between food-related quality of life, subjective olfactory function, and duration of olfactory loss using the REML method.

## 3. Results

### 3.1. Demographics and Olfactory Deficits

In total, 688 participants completed part 1 + 2 (*n* = 156) or part 1 + 2 + 3 (*n* = 532) of the questionnaires, of which 166 were normosmic controls. Among the 522 participants with olfactory dysfunction, 251 participants reported experiencing parosmia often (*n* = 131) or always (*n* = 120) in addition to their olfactory loss (olfactory distortion group) while 271 participants reported experiencing parosmia rarely (*n* = 110) or never (*n* = 161) in addition to their olfactory loss (olfactory loss group), see Table 1. Subjective olfactory function was weakly negatively correlated with the duration of smell loss (see Figure 1). As expected, participants with smell loss due to COVID-19 had a shorter duration of smell loss (mean: 73 months, *p* < 0.0001) and were younger (mean 13.5 years, *p* < 0.0001) than non-COVID participants. 

### 3.2. Quality of Life

The ratings of current food-related quality of life were significantly different between the normosmic group (mean 82.53; 95% CI (78.56; 86.48)) and the combined olfactory groups (mean 62.54; 95% CI (60.32; 64.77)); (F(1,671) = 74.5896, *p* = <0.0001). Within the olfactory groups, the current food-related quality of life was not found to be significantly different between the rated severity of parosmia (no/rare/often/always) (F(3,510) = 0.1542, *p* = 0.9270). 

Similarly, the loss in food-related quality of life from before the olfactory dysfunction to the current rating was not significantly different between the olfactory loss group and the parosmia group (mean difference 0.63 (95% CI (−4.8781; 6.1376); *p* = 0.8223).

Food-related quality of life was weakly correlated to severity of subjective olfactory function (see Figure 1).

### 3.3. Weight Changes

Overall, 48.5% of participants had experienced weight change due their olfactory dysfunction, where 22.7% had experienced a decrease in weight, 16.3% an increase in weight, and 9.5% more fluctuations in weight. 

Weight change was significantly correlated with parosmia (χ2 = 38.09, *p* < 0.0001), where, notably, 39.2% of participants with constant parosmia (‘always distorted smells’) reported a decrease in weight (see Figure 2). 

There was a significant difference between genders (χ2 = 15.792, *p* = 0.0013), where male participants were more likely to experience either no change in weight or weight loss. Among the male participants with constant parosmia, 53.0% reported weight loss as compared with 36.9% of female participants. 

Participants with a longer duration of parosmia (non-COVID participants) had a similar frequency of weight gain, but a lower frequency of reported weight loss and fluctuations (χ2 = 13.46, *p =* 0.0037). 

### 3.4. Changed Awareness of Other Senses Following Olfactory Dysfunction

While participants rarely reported increased or decreased awareness of vision, hearing, or touch (hands) following their olfactory dysfunction, 45.1% reported higher awareness of mouth feel. 

For taste, only 26.5% reported no change in awareness, while 43.6% reported higher awareness after olfactory dysfunction. The relatively large number of participants reporting lower taste awareness (29.9%) was associated with decreased subjective taste function (χ2 = 60.83, *p* < 0.0001), as the only 17.0% of participants with a subjective normal quantitative sense of taste (subjective taste function of VAS > 90/100) and 21.3% of participants without any history of dysgeusia reported lower taste awareness (see Figure 3). 

Some gender differences were found, as male participants more frequently reported no change in sensory awareness for both vision (85.3% vs. 76.6%, χ2 = 7.6, *p* = 0.1074), hearing (83.3% vs. 78.1%, χ2 = 3.777, *p* = 0.4370), touch (hands) (90.1% vs. 84.0%, χ2 = 4.935, *p* = 0.2940), mouthfeel (60.8% vs. 41.1%, χ2 = 14.126, *p* = 0.0069), and taste (44.6% vs. 22.0% χ2 = 23.113, *p* = 0.0001). 

Participants with non-COVID aetiologies (longer duration) also differed some aspects of reported changes in sensory awareness: vision (increase: 17.3% vs. 11.1%; decrease: 8.8% vs. 7.8%, χ2 = 7.6, *p* = 0.1074), hearing (increase: 14.2% vs. 12.4%, decrease: 6.1% vs. 8.82%, χ2 = 5.122, *p* = 0.2750), touch (hands) (increase: 11.7% vs. 10.1%, decrease: 5.6% vs. 3.0%, χ2 = 4.247, *p* = 0.3737), mouthfeel (increase: 45.2% vs. 45.0%, decrease: 7.0% vs 11.7%, χ2 = 6.350, *p* = 0.1745), and taste (increase: 44.6% vs. 22.0% decrease: 22.3% vs. 34.7%, χ2 = 23.113, *p* = 0.0001). 

### 3.5. Cooking and Food Habits

#### 3.5.1. Differences between Participants with Normosmia and Olfactory Dysfunction

The total CAFPAS score was significantly differed between the normosmic participants (mean 13.03; 95% CI (12.70; 13.70)) and participants with olfactory dysfunction (mean 12.11; 95% CI (11.89; 12.34)) (F(1,531) = 19.9621, *p* < 0.0001). These two groups also differed in the CAFPAS Skill and Self-Efficacy subscore (F(1,531) = 17.5546, *p* < 0.0001), and CAFPAS Attitude subscore (F(1,531) = 29.4597, *p* < 0.0001), but not in the CAFPAS Structure factors subscore (F(1,531) = 0.2427, *p* = 0.6225). 

No significant difference in the total CAFPAS score was found between gender (mean 0.31; 95% CI (−0.29; 0.91), *p* = 0.3104) or aetiology (COVID vs non-COVID (long duration)) (mean 0.17; 95% CI(−0.29; 0.63), *p* = 0.4631).

#### 3.5.2. Parosmia Severity

Parosmia severity was not found to be associated with differences in total CAFPAS score (F(3,365) = 0.7342, *p* = 0.5322), CAFPAS Skill and Self-Efficacy subscore (F(3,365) = 1.0006, *p* = 0.3926), or CAFPAS Attitude subscore (F(3,365) = 1.1516, *p* = 0.3282). 

In the CAFPAS Structure factors subscore, a significant difference was found between participants with increasing parosmia severity (F(3,365) = 2.888, *p* = 0.0355) (No parosmia (mean 3.70; 95% CI (3.52; 3.88), rare (mean 3.52; 95% CI (3.30; 3.75)), often (mean 3.37; 95% CI (3.17; 3.58)), always (mean 3.33; 95% CI (3.12; 3.54))). 

For differences in individual items of the questionnaire, see Table 2. Note that participants with olfactory disorders especially want to get through cooking as soon as possible, feel cooking is less fulfilling, are less comfortable preparing food, feel less inspired to cook for other people, and find it more difficult to accomplish desired results during cooking. Additionally, for parosmic participants, there was a lower ability to decide what to eat, a lower confidence in the ability to deal with unexpected results during cooking, and a wish for more time to plan meals. 

### 3.6. Food Item Recognition, Liking, and Frequency of Intake

#### 3.6.1. Recognition

Participants with olfactory dysfunction were significantly less able to recognise all categories of food items during consumption using taste, smell, and mouthfeel combined.

Generally, the ability to recognise items was comparable between participants with olfactory dysfunction, however participants with parosmia were significantly worse than participants with olfactory loss in recognising several food items: Coffee (*p* = 0.0031); Chocolate (*p* = 0.0147); menthol (*p* = 0.0134); cucumber (*p* = 0.0474); watermelon (*p* = 0.0471); granola (*p* = 0.0314); crisps (*p* = 0.0083), red wine (*p* = 0.0054), and white wine (*p* = 0.0426). Participants with parosmia were only superior to participants with olfactory loss in recognition of cinnamon (*p* = 0.0162) (see Table 3).

#### 3.6.2. Liking

Apart from the pure gustatory food items (refined sugar and table salt), liking scores of food items were lower in participants with olfactory dysfunction (combined groups) compared with normosmic controls (see Table 3). 

For the olfactory loss group, liking of barbeque sauce was similar to controls while other food items were rated with lower liking scores. For ranking of differences in liking between controls and the olfactory loss group, see Figure 4. Differences in ranking of liking differences were found in the parosmia group compared with controls, see Figure 5.

Although there was a tendency of higher linking of strong food items among participants with increased mouthfeel sensory awareness, this was not significant for any of the food items (Black pepper: (F(4,499) = 1.7014, *p* = 0.1484); Chili: (F(4,499) = 1.0445, *p* = 0.3836); Menthol: (F(4,499) = 0.1041, *p* = 0.9811); Ginger: (F(4,499)=0.6898, *p* = 0.5993); Mustard: (F(4,499) = 0.5026, *p* = 0.7339); Wasabi: (F(4,499) = 1.6418, *p* = 0.1625)).

#### 3.6.3. Frequency of Intake

In spite of numerous food items with significantly lower ratings of both recognisability and liking, many food items were not significantly less frequently used in participants with olfactory loss and olfactory distortions. This may reflect that participants do not always cook their own meals, but may also disclose a pattern of following generic recipes, recipes previously used before the occurrence of the olfactory disorder, or that recipes were made to meet expectations of others without olfactory disorders sharing the meal. Nonetheless, it reveals a potential need for further investigations of new pleasurable food items and recipes, and a potential problem as patients with olfactory disorders have been shown to have a higher level of food neophobia, and a lower willingness to try new food [15].

## 4. Discussion

We found that olfactory dysfunction was associated with a lower food-related quality of life, as previously documented in the literature. However, higher severity of parosmia was neither associated with a lower current rating of food-related quality of life nor a larger loss in quality of life. As such, we could not find a difference in quality-of-life impact between mild and severe parosmia. Nonetheless, the severity of parosmia was significantly associated with weight loss. Cooking habits were found to have significant differences between participants with olfactory dysfunction and normosmic controls. Similarly, distribution of food item recognition, liking, and frequency of intake differed significantly between groups.

### 4.1. Cooking Habits

From the Cooking and Food Provisioning Action Scale, we found that normosmic participants differed significantly from participants with olfactory dysfunction in both total scores, in the Skill and Self-Efficacy and attitude subscores, and on multiple individual items (see Table 2). Most pronounced was the wish to get through with cooking as soon as possible, the reduced attitude towards cooking as a fulfilling activity, the lack of comfort in preparing food, the lack of inspiration to cook for others, the feeling that cooking was a waste of effort, and the inability to make new foods successfully. 

By comparing the olfactory loss group with the parosmia group, the results were strikingly similar apart from a reduced ability to decide what to eat, a reduced confidence in the ability to deal with unexpected results, and a wish for more time for planning meals in the parosmia group. With the common combination of both olfactory loss and parosmia, preparing a meal that is both multisensory stimulating, securely prepared, and varied does require time and thought. These differences may reflect a more demanding task of finding pleasurable foods and preparing a meal that avoids having parosmic triggers. As described by Parker et al., the molecular components of parosmic triggers are widespread and can emerge from both the ingredients and preparation methods of cooking, especially heating and roasting [10]. 

### 4.2. Food-Related Quality of Life and Weight Changes in Olfactory Disorders

Food-related quality of life was significantly reduced in participants with olfactory dysfunction; however, parosmia was not associated with further a decrease in subjective food-related quality of life. While we found a weak correlation between food-related quality of life and subjective olfactory function, no mentionable effect was found for the duration of olfactory dysfunction, indicating little or no regaining of food-related quality of life over time. One explanation could be, that participants change food preferences and eating behaviour after their olfactory dysfunction [8], although there is contradictory evidence in the literature [16]. We found that while the frequency of intake was less affected than liking (see Table 3), there were large differences in frequency of intake between individual food items and between participants with and without severe parosmia. As such, the heterogeneous patient population suffering from olfactory dysfunction is highly relevant to consider when assessing differences in preference and eating patterns. This is furthermore emphasised by the significant differences in weight change, where weight loss was more pronounced in participants suffering from parosmia.

It is well known that olfactory dysfunction is associated with both weight gain and weight loss in patients with hyposmia and anosmia [17]. Several studies have highlighted patterns of unbalanced diets in patients with olfactory dysfunction, although these findings may reflect changes in intake and preference for specific items and not general patterns [18]. Moreover, obesity is also associated with lower olfactory test scores indicating that the causality of underlying mechanisms may be more complex [19]. 

The current findings that the occurrence of weight loss is significantly higher in participants with parosmia compared with olfactory loss highlights the need for increased focus on dissimilar, though tailored support measures for patients suffering from different subtypes of olfactory dysfunction.

### 4.3. Sensory Awareness

Following the onset of olfactory dysfunction, a high number of participants reported increased awareness of mouthfeel (45.1%) and basic taste (43.6%), while 29.9% reported decreased taste awareness. This decreased taste awareness can be partly attributed to the cooccurrence of quantitative taste loss or qualitative taste dysfunction which is a well-described attribute in especially post-COVID olfactory loss [20]. However, it may also be influenced by the common inability to distinguish taste from retronasal olfaction [21]. COVID-related anosmia was sometimes accompanied by changes in trigeminal nerve sensitivity, with people becoming hypo-sensitive to chemesthesis and fewer becoming hypersensitive to its effects, miming a difference in perceived mouthfeel of foods. There is little discussion of these changes in parosmia. When comparing differences in liking of strong foods (trigeminal stimulants) between normosmic controls and parosmic participants, the greatest divergence is with menthol. However, there is strong evidence that mint is a trigger of unpleasant distortions common to parosmia [22].

### 4.4. Food Items

The included list of food items was selected to include all basic tastants and different mouthfeel stimuli in order to enable patients and chefs to construct a broad multisensory stimulating meal in order to compensate for olfactory dysfunction. The distribution of altered liking of these food items differed depending on the nature of the olfactory dysfunction. This is illustrate ranked linking scores compared with normosmic controls and can give supportive information (to, e.g., patients and chefs) on potentially safer food items for patients with olfactory loss (Figure 4) and parosmia (Figure 5), respectively.

In the parosmia group, disliked food items included known parosmic triggers such as coffee, chocolate, cucumber, and toasted bread [22]. While the previous studies by Parker et al. have had a broad focus to investigate underlying molecular mechanisms of parosmia-related receptor binding and parosmic triggers within a broad range of domains (personal care, home and environment, food, and beverages) [10,22], the current focus investigates both aspects of the hedonic range of food items in both olfactory loss and parosmia for food items that can be used to stimulate basic tastants and mouthfeel. 

The use of condiments to increase the palatability of prepared meals is a documented strategy for patients with the olfactory loss [8], however, by increasing focus on all the ingredients in olfactory-deficit-specific recipes, a higher degree of eating diversity may be achieved. As such, our findings supplement the existing knowledge of the hedonic yield of food in patients with olfactory dysfunction. The general importance of a well-balanced diet has been proposed as a dietary approach for treating post-COVID olfactory loss [23], while beneficial effects on olfactory function of omega-3 fatty acids have been described in both patients following rhino surgery and in patients with post-viral olfactory dysfunction [24,25]. We propose increased focus on multisensory stimulation of taste and mouthfeel by including generally well-liked ingredients for optimising enjoyment of food and diet in both olfactory loss and parosmia, including contrasts of textures and temperature of food ingredients in meals.

### 4.5. Gender Differences

As seen in previous patient populations with olfactory dysfunction [26], we found a greater proportion of women among participants with olfactory dysfunction. While the female gender has been shown to be associated with slightly superior olfactory function in a larger study population [27], the gender-related differences in consequences following parosmia have not been well established. No gender differences were found in the CAFPAS score. Following parosmia, males were more likely to experience either no change in weight or weight loss. Furthermore, male participants more frequently reported no change in sensory awareness for all senses, however, only statistically significant for taste and mouthfeel. 

### 4.6. Limitations

The current study was based on a questionnaire without testing olfactory function or stimulation of food items. Currently, only one method of quantitatively assessing the severity of parosmia has been published, however, the SSParoT has currently only been validated in a normative population [28]. As such, no clinically validated test of parosmia severity exists for clinical use. The current findings on subjective ratings of parosmia intensity and does not differentiate between food and non-food parosmic triggers. These findings should be repeated in a future study when an appropriate test for evaluating parosmia has been validated. Similarly, weight changes were asses by asking participants to characterise and quantify weight changes due to olfactory dysfunction. 

Due to the length of the questionnaire, not all potential factors affecting cooking habits were included in the data collection and analysis. We focused on having an age-matched control group with more females to ensure that the core demographics were comparable. Further studies on cooking habits are needed to explore potential effects of, e.g., number of persons in the household, children, hours of weekly work, and other aspects of social context.

Participants were included with different etiologies, as the non-COVID participants had a longer duration of parosmia which made it possible to assess more long-term effects of parosmia on weight changes, sensory awareness, and the CAFPAS scores. While earlier investigations on parosmic patients, post-COVID and pre-COVID parosmia were found to be similar in TDI-scores and gas chromatography-olfactometry scores [10], it cannot be determined if the current findings on differences in weight changes and sensory awareness are due to the longer duration of parosmia or if non-COVID parosmia differs from post-COVID parosmia.

Differences in food item scales between groups may differ. However, for items less used in non-processed foods such as palm oil, only minor differences were found between groups, indicating some degree of alignment. 

During the design of the study, food items were selected based on taste and mouthfeel potential. However, for patients with parosmia, the old saying that it only takes one bad fish to ruin a bouillabaisse is most likely very accurate. With the limited number of food items that could be included in a questionnaire, and the heterogenic distribution of parosmic triggers, the recommended items are not likely to be representative of all patients with parosmia.

Some findings of the study were hypothesis building and can give patients, chefs, and researchers inspiration for perusing increased liking of meals in homes, professional kitchens, and research settings.

## 5. Conclusions

There are several possible avenues for increasing the pleasure of food in patients with olfactory dysfunction if sufficient attention is paid to the type of olfactory dysfunction. While the Cooking and Food Provisioning Action Scale provided insights of obstacles in the ability to prepare meals for participants with olfactory loss and parosmia, the differences in liking also provides insights to which ingredients are more likely to create a pleasurable meal in olfactory loss and parosmia, respectively, with ‘safer food items’. As there is an increased sensory awareness of taste and mouthfeel in these patients, an increased focus on diversity in basic tastant stimulation and trigeminal stimulation of mouthfeel can increase the multisensory stimulation of a meal using commonly tolerated ingredients. More research is needed to test the effects of targeted recipes for different types of olfactory dysfunction on cooking habits, diet, and quality of life. 

## Figures and Tables

**Figure 1 foods-11-01686-f001:**
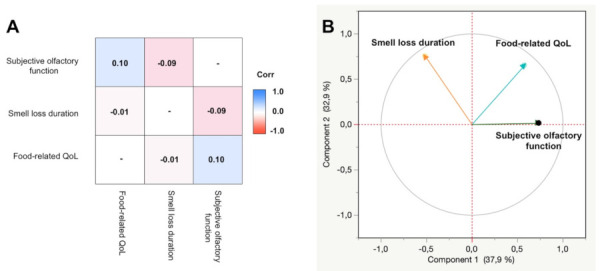
Correlations between the three principal components with respect to the food-related quality of life, duration of smell loss, and subjective olfactory function. (**A**) Correlation matrix. Shades of blue indica ate positive correlation, whereas shades of red indicate negative correlations. (**B**) Correlation circle of principal component analysis.

**Figure 2 foods-11-01686-f002:**
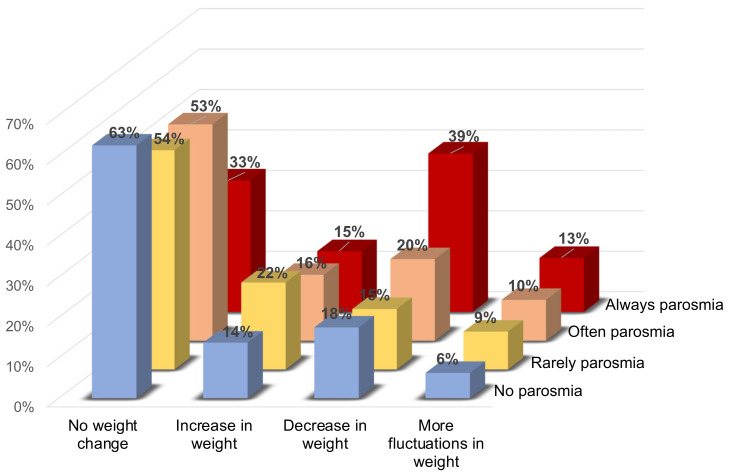
Weight change in parosmia. The incidence of weight changes is compared between participants with no parosmia (blue), rarely parosmia (yellow), often parosmia (orange), and always parosmia (red). Higher severity of parosmia is associated with a higher incidence of weight loss, more fluctuations in weight, and less stabile weight (no weight change).

**Figure 3 foods-11-01686-f003:**
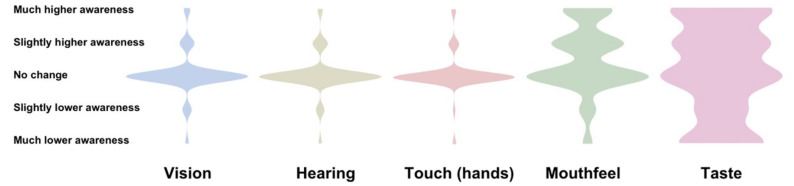
Changed awareness of other senses following olfactory dysfunction.

**Figure 4 foods-11-01686-f004:**
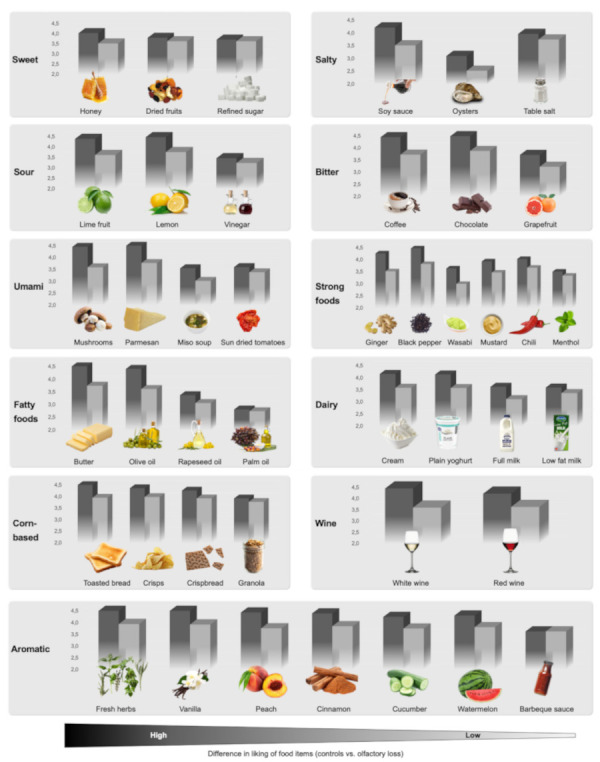
Liking of food items in olfactory loss (grey) compared with normosmic controls (black). For all food items categories (basic tastes, fatty, dairy, corn-based, wine, and aromatic foods) food items are listed from left to right according to the difference in liking between normosmic controls and participants with olfactory loss (high to low difference). When choosing ingredients, food items on the right side of each category are more similar in liking between participants with olfactory loss and normosmic controls. This may indicate ‘safer foods’, however, please see Table 3, as some items are rated lower in the frequency of intake, which may influence ratings of liking.

**Figure 5 foods-11-01686-f005:**
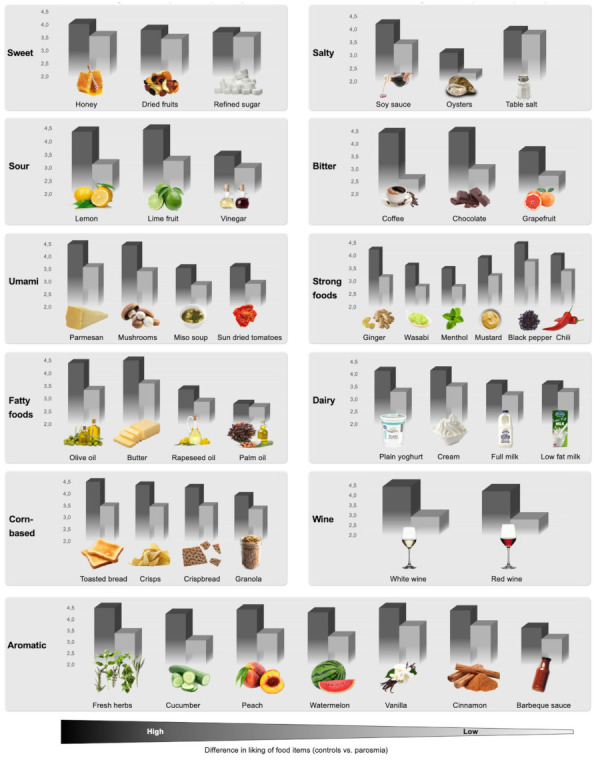
Liking of food items in parosmia (grey) compared with normosmic controls (black). For all food items categories (basic tastes, fatty, dairy, corn-based, wine, and aromatic foods) food items are listed from left to right according to the difference in liking between normosmic controls and parosmic participants (high to low difference). When choosing ingredients, food items on the right side of each category are more similar in liking between parosmic participants and normosmic controls. This may indicate ‘safer foods’, however, please see Table 3, as some items are rated lower in the frequency of intake, which may influence ratings of liking.

**Table 1 foods-11-01686-t001:** Demographics and olfactory disorders. Inter quartile ranges were added to give an overview of the distribution of data, which reflects that the duration of smell loss did not follow parametric distribution as some patients had suffered from olfactory deficits for several years. IQR: Inter-quartile range; QoL: Quality of life.

	Olfactory Dysfunction (*n* = 522)	Normosmic Controls (*n* = 166)	Olfactory Loss (*n* = 271)	Parosmia (*n* = 251)
Age (mean, IQR)	47 (34–58)	47 (37–58)	51 (40–63)	42 (30–53)
Sex, female (*n*, %)	417 (80%)	111 (67%)	200 (74%)	217 (86%)
Country of residence (*n*, %) - Denmark - United Kingdom - United States - Other	213 (41%) 98 (19%) 129 (25%) 82 (16%)	85 (51%) 72 (43%) 2 (1%) 7 (5%)	142 (53%) 36 (13%) 59 (22%) 34 (12%)	71 (29%) 62 (25%) 70 (29% 48 (19%)
Aetiology (*n*, %) - COVID-19 - Post-viral (non-COVID) - Sino-nasal disease - Head trauma - Other - Don’t know	319 (61%) 48 (9%) 24 (5%) 25 (5%) 20 (4%) 85 (16%)	-	121 (45%) 32 (12%) 22 (8%) 19 (7%) 16 (6%) 60 (22%)	(79%) (6%) (1%) (2%) (2%) (10%)
Duration (months; mean, IQR)	35 (5–24)	-	54 (4–48)	15 (5–10)
Subjective olfactory function (VAS 0-100, IQR)	29 (6–50)	100 (100–100)	22 (1–32)	37 (19–50)
Parosmia (*n*, %) - Never - Rarely - Often - Always	161 (31%) 110 (21%) 131 (25%) 120 (23%)		161 (59%) 110 (41%) - -	- - 131 (52%) 120 (48%)
Weight chang - No change - Weight loss - Weight gain - More fluctuations	266 (52%) 117 (23%) 84 (16%) 49 (9%)		158 (59%) 44 (16%) 45 (17%) 20 (7%)	108 (43%) 73 (29%) 39 (16%) 29 (12%)
Food-related QoL (mean, IQR)	63 (41–86)		63 (41–84)	62 (41–87)

**Table 2 foods-11-01686-t002:** Differences in cooking and food provisioning habits between normosmics and patients with olfactory disorders. Listed according to differences between normosmic controls and olfactory dysfunction, in descending order, where the largest differences are at the top. Mean values (95% CI) on a 1-7 scale. Δ = difference; asap = as soon as possible.

	Normosmic (*n* = 166)	Olfactory Dysfunction (*n* = 367)	Δ	*p*-Value	Olfactory Loss (*n* = 191)	Parosmia (*n* = 176)	Δ	*p*-Value
Want to get through cooking asap.	2.58 (2.28; 2.88)	3.85 (3.65; 4.06)	1.27	<0.0001	3.97 (3.68; 4.27)	3.72 (3.41; 4.02)	0.25	0.2359
Cooking is a fulfilling activity	5.70 (5.42; 5.98)	4.74 (4.55; 4.93)	0.96	<0.0001	4.75 (4.47; 5.03)	4.72 (4.43; 5.01)	0.03	0.8692
Comfortable preparing food	6.21 (5.95; 6.47)	5.28 (5.10; 5.45)	0.93	<0.0001	5.39 (5.12; 5.65)	5.15 (4.88; 5.44)	0.24	0.2540
Inspired to cook for other people	5.67 (5.39; 5.96)	4.76 (4.56; 4.95)	0.91	<0.0001	4.74 (4.46; 5.02)	4.78 (4.48; 5.07)	0.04	0.8565
Easy to accomplish desired results during cooking	5.67 (5.42; 5.93)	4.78 (4.61; 4.95)	0.89	<0.0001	4.86 (4.60; 5.11)	4.70 (4.44; 4.97)	0.15	0.4137
Cooking is a waste of effort	1.89 (1.64; 2.15)	2.74 (2.57; 2.92)	0.85	<0.0001	2.79 (2.54; 3.05)	2.69 (2.42; 2.96)	0.10	0.5746
Inability to make new foods successfully	3.43 (3.16; 3.71)	4.22 (4.04; 4.41)	0.79	<0.0001	4.40 (4.14; 4.66)	4.03 (3.76; 4.30)	0.37	0.0538
Cooking brings little enjoyment	3.31 (3.02; 3.61)	4.02 (3.82; 4.22)	0.71	<0.0001	4.08 (3.81; 4.35)	3.95 (3.66; 4.23)	0.13	0.5136
Confidence in ability to deal with unexpected results during cooking	5.38 (5.11; 5.64)	4.72 (4.54; 4.90)	0.66	<0.0001	4.93 (4.68; 5.19)	4.49 (4.23; 4.76)	0.44	0.0209
Ability to decide what to eat	4.99 (4.70; 5.28)	4.38 (4.18; 4.57)	0.62	0.0006	4.63 (4.34; 4.91)	4.11 (3.81; 4.40)	0.52	0.0131
Cooking for others is a burden	2.57 (2.28; 2.85)	3.19 (3.00; 3.39)	0.62	0.0004	3.18 (2.90; 3.45)	3.21 (2.92; 3.50)	0.03	0.8838
Coping with problems during cooking	5.67 (5.43; 5.92)	5.07 (4.90; 5.23)	0.61	<0.0001	5.15 (4.90; 5.39)	4.98 (4.73; 5.24)	0.17	0.3651
Prefer to spend time on more important things than cooking	3.20 (2.92; 3.47)	3.73 (3.54; 3.91)	0.53	0.0020	3.74 (3.49; 4.00)	3.70 (3.43; 3.98)	0.04	0.8299
Limited by lack of cooking knowledge	2.36 (2.07; 2.65)	2.78 (2.58; 2.97)	0.42	0.0195	2.80 (2.51; 3.08)	2.76 (2.46; 3.06)	0.04	0.8709
Involvement in daily meal preparations	6.15 (5.88; 6.42)	5.73 (5.55; 5.91)	0.42	0.0109	5.87 (5.61; 6.14)	5.58 (5.31; 5.85)	0.29	0.1270
Prefer to cook than having food prepared	4.31 (4.01; 4.61)	3.95 (3.75; 4.15)	0.36	0.0543	3.91 (3.63; 4.20)	3.99 (3.69; 4.28)	0.08	0.7276
Confidence in creating meals from ingredients on hand	6.04 (5.81; 6.28)	5.80 (5.65; 5.96)	0.24	0.0982	5.88 (5.66; 6.10)	5.72 (5.49; 5.96)	0.16	0.3430
Confidence in choosing between similar products	5.93 (5.71; 6.16)	5.74 (5.59; 5.89)	0.19	0.1543	5.76 (5.54; 5.98)	5.71 (5.49; 5.94)	0.05	0.7644
Reflection on what to cook and eat	5.14 (4.88; 5.41)	5.00 (4.83; 5.18)	0.14	0.3765	4.84 (4.59; 5.09)	5.18 (4.92; 5.44)	0.34	0.0675
Knowledge of usage of ingredients during purchasing	6.17 (5.97; 6.38)	6.04 (5.90; 6.18)	0.13	0.2915	6.04 (5.84; 6.25)	6.04 (5.83; 6.25)	0.00	0.9888
Difficult finding time to prepare preferred food	3.63 (3.35; 3.90)	3.76 (3.57; 3.94)	0.13	0.4452	3.54 (3.29; 3.79)	3.99 (3.73; 4.26)	0.45	0.0154
Knowledge of where to find needed ingredients	6.34 (6.15; 6.53)	6.23 (6.01; 6.35)	0.11	0.3138	6.19 (6.01; 6.38)	6.26 (6.07; 6.46)	0.07	0.5953
No time to prepare meals due to family responsibilities	2.62 (2.36; 2.88)	2.71 (2.54; 2.89)	0.09	0.5595	2.67 (2.43; 2.92)	2.76 (2.51; 3.01)	0.09	0.6338
No time to prepare meals due to job responsibilities	3.31 (3.00; 3.62)	3.24 (3.03; 3.45)	0.07	0.7109	2.93 (2.64; 3.21)	3.58 (3.28; 3.88)	0.65	0.0022
Knowledge of kitchen equipment usage	6.37 (6.19; 6.56)	6.33 (6.20; 6.45)	0.04	0.6777	6.27 (6.10; 6.45)	6.39 (6.21; 6.57)	0.11	0.3639
Mental plan of steps before cooking	5.82 (5.59; 6.05)	5.78 (5.62; 5.94)	0.04	0.7814	5.80 (5.57; 6.02)	5.76 (5.52; 5.99)	0.04	0.8073
No time to prepare meals due to social responsibilities	2.43 (2.19; 2.66)	2.47 (2.31; 2.63)	0.04	0.7780	2.36 (2.15; 2.58)	2.58 (2.36; 2.81)	0.22	0.1717
Wish for more time to plan meals	3.98 (3.73; 4.24)	4.01 (3.84; 4.18)	0.03	0.8541	3.77 (3.53; 4.01)	4.28 (4.03; 4.52)	0.51	0.0040

**Table 3 foods-11-01686-t003:** Recognition, liking, and frequency of intake for food items and taste categories in normosmic controls, patients with olfactory loss (OL), and patients with parosmia. Recognition was rated by how likely the participant was to recognise the food item during consumption using taste, smell, and mouthfeel combined. Liking was rated for the current liking of food items ranging from disgust to pleasure. Ratings of recognition, liking, and frequency of intake ranged from 1–5, where 1 indicated lowest possible and 5 indicated highest possible. Control: Normosmic controls; OL: Olfactory loss group. * indicates general scores of the basic tastant (asked without mentioning specific food items).

	Recognition				Liking				Frequency of Intake			
	Control	OL	Parosmia	*p*-value	Control	OL	Parosmia	*p*-value	Control	OL	Parosmia	*p*-value
**Sweet food items**												
Sweet foods *	4.94	4.08	4.07	<0.0001	4.37	4.16	4.06	0.0118	4.01	3.89	3.85	0.3121
Refined sugar	4.43	3.64	3.62	<0.0001	3.71	3.63	3.53	0.2685	3.91	3.86	3.78	0.6250
Dried fruits	4.78	3.45	3.27	<0.0001	3.80	3.63	3.45	0.0137	3.41	3.31	3.25	0.4866
Honey	4.78	3.43	3.49	<0.0001	4.03	3.52	3.56	<0.0001	3.21	3.07	3.09	0.5579
**Salty food items**												
Salty foods *	4.95	4.12	4.14	<0.0001	4.31	4.05	4.08	0.0257	3.92	3.67	3.64	0.0340
Soy sauce	4.69	3.26	3.24	<0.0001	4.21	3.51	3.44	<0.0001	3.52	2.98	3.05	<0.0001
Table salt	4.69	3.94	4.00	<0.0001	3.96	3.74	3.81	0.0994	4.57	4.42	4.26	0.0168
Oysters	3.87	2.79	2.65	<0.0001	3.10	2.52	2.33	<0.0001	1.63	1.35	1.48	0.0018
**Sour food items**												
Sour foods *	4.86	3.85	3.81	<0.0001	4.00	3.67	3.31	<0.0001	4.26	3.94	3.53	<0.0001
Vinegar	4.87	3.57	3.54	<0.0001	3.46	3.23	3.00	0.0016	3.42	3.04	2.92	0.0009
Lemon	4.89	3.69	3.49	<0.0001	4.46	3.75	3.27	<0.0001	4.02	3.54	3.28	<0.0001
Lime	4.62	3.47	3.37	<0.0001	4.38	3.61	3.14	<0.0001	3.38	3.00	2.91	0.0006
**Bitter food items**												
Bitter foods *	4.88	3.85	3.62	<0.0001	3.44	2.94	2.80	<0.0001	3.87	3.35	3.13	<0.0001
Grapefruit	4.62	3.27	3.01	<0.0001	3.71	3.22	2.74	<0.0001	2.32	2.06	2.00	0.0177
Coffee	4.94	3.26	2.88	<0.0001	4.44	3.72	2.60	<0.0001	4.39	4.08	3.57	<0.0001
Chocolate	4.90	3.45	3.15	<0.0001	4.48	3.87	3.00	<0.0001	3.69	3.34	2.95	<0.0001
**Umami-rich food items**												
Umami foods *	3.87	3.20	3.05	<0.0001	4.17	3.47	3.16	<0.0001	4.03	3.45	3.29	<0.0001
Mushrooms	4.61	3.09	3.05	<0.0001	4.45	3.59	3.43	<0.0001	3.46	2.95	3.00	0.0001
Parmesan cheese	4.68	3.18	3.16	<0.0001	4.63	3.77	3.59	<0.0001	3.55	3.08	3.10	0.0001
Sun-dried tomato	4.62	3.01	2.83	<0.0001	3.60	3.38	2.93	<0.0001	2.62	2.41	2.24	0.0034
Miso soup	3.61	2.73	2.74	<0.0001	3.55	3.02	2.88	<0.0001	1.96	1.66	1.77	0.0203
**Strong food items**												
Strong foods *	4.94	3.87	3.77	<0.0001	4.15	3.77	3.55	<0.0001	3.79	3.59	3.46	0.0269
Black pepper	4.77	3.65	3.61	<0.0001	4.46	3.81	3.78	<0.0001	4.62	4.32	4.11	<0.0001
Chili	4.80	3.73	3.61	<0.0001	4.02	3.66	3.40	<0.0001	3.65	3.29	3.19	0.0017
Menthol	4.64	3.52	3.23	<0.0001	3.50	3.33	2.80	<0.0001	2.32	2.36	2.36	0.9629
Ginger	4.87	3.38	3.27	<0.0001	4.25	3.51	3.18	<0.0001	3.44	2.98	2.90	0.0001
Mustard	4.76	3.50	3.27	<0.0001	3.92	3.46	3.22	<0.0001	3.19	3.00	2.82	0.0112
Wasabi	4.42	3.40	3.19	<0.0001	3.63	2.99	2.81	<0.0001	2.31	1.96	1.95	0.0046
	**Recognition**				**Liking**				**Frequency of intake**			
	Control	OL	Parosmia	*p*-value	Control	OL	Parosmia	*p*-value	Control	OL	Parosmia	*p*-value
**Fatty food items**												
Olive oil	4.54	2.92	2.89	<0.0001	4.40	3.61	3.35	<0.0001	4.49	3.84	3.75	<0.0001
Butter	4.69	3.16	3.21	<0.0001	4.54	3.74	3.60	<0.0001	4.37	4.16	3.94	0.0016
Rapeseed oil	3.39	2.58	2.42	<0.0001	3.37	3.06	2.89	<0.0001	3.13	2.78	2.44	<0.0001
Palm oil	2.68	2.45	2.34	0.0229	2.81	2.74	2.68	0.3920	1.70	1.61	1.56	0.4235
**Dairy food items**												
Cream	4.54	3.17	3.19	<0.0001	4.15	3.58	3.51	<0.0001	3.27	3.16	2.89	0.0023
Milk (low fat)	4.19	3.04	3.02	<0.0001	3.59	3.36	3.29	0.0666	3.63	3.39	3.24	0.1105
Milk (full)	4.31	3.01	3.13	<0.0001	3.62	3.11	3.16	0.0002	2.72	2.38	2.37	0.0568
Yoghurt (plain)	4.55	3.26	3.26	<0.0001	4.13	3.57	3.30	<0.0001	3.36	3.19	3.13	0.3014
**Aromatic food items**												
Fresh herbs	4.52	2.96	2.91	<0.0001	4.66	3.94	3.39	<0.0001	4.37	3.97	3.69	<0.0001
Vanilla	4.82	3.02	3.21	<0.0001	4.59	3.92	3.71	<0.0001	3.23	3.09	3.13	0.4004
Cinnamon	4.88	3.14	3.44	<0.0001	4.39	3.85	3.73	<0.0001	3.18	3.06	3.09	0.5841
Barbeque sauce	4.31	3.17	3.02	<0.0001	3.63	3.63	3.14	<0.0001	2.26	2.58	2.42	0.0194
Cucumber	4.79	3.33	3.07	<0.0001	4.24	3.75	3.08	<0.0001	3.82	3.51	3.01	<0.0001
Watermelon	4.74	3.35	3.10	<0.0001	4.30	3.81	3.25	<0.0001	2.56	2.50	2.40	0.2221
Peach	4.65	3.20	3.05	<0.0001	4.43	3.76	3.38	<0.0001	2.45	2.38	2.29	0.2922
**Corn-based food items**												
Bread (toasted)	4.80	3.42	3.24	<0.0001	4.56	3.95	3.48	<0.0001	3.72	3.56	3.52	0.2555
Crisps	4.80	3.58	3.25	<0.0001	4.36	3.98	3.46	<0.0001	3.14	3.25	3.20	0.6423
Crispbread	4.67	3.41	3.17	<0.0001	4.26	3.92	3.49	<0.0001	3.35	3.38	3.21	0.2230
Granola	4.44	3.32	3.06	<0.0001	3.92	3.77	3.35	<0.0001	2.89	2.99	2.88	0.6201
**Wine**												
Red wine	4.77	3.30	2.95	<0.0001	4.21	3.62	2.80	<0.0001	3.13	2.88	2.55	0.0002
White wine	4.68	3.19	2.94	<0.0001	4.44	3.59	2.93	<0.0001	3.38	2.77	2.55	<0.0001

## Data Availability

The data supporting the findings of this study are available from the corresponding author upon reasonable request.

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
