# Peer review of "The Effects of Olfactory Loss and Parosmia on Food and Cooking Habits, Sensory Awareness, and Quality of Life—A Possible Avenue for Regaining Enjoyment of Food"

_foods, 2022, doi:10.3390/foods11121686_

Round 1

Reviewer 1 Report

In this MS, using an online self-assessment testing, Fjaeldstad and Smith interestingly questioned the fate of food and cooking habits of parosmic and anosmic subjects. The protocol is the Cooking and Food Provisioning Action Scale. This questionnaire also probes recognition, liking and frequency of food intake by categories. The study underlies the fact that parosmia and anosmia are associated with comfort and inspiration for cooking. Parosmia was shown to be associated with a higher incidence of weight loss.

1. Gender

Table 1 clearly shows that 80% of the 522 subjects are women, thus the results of this study present a massive bias towards female odor perception. Surprisingly, this fact is not discussed at any time in the MS. It has to be, since olfactory threshold, discrimination and identification are different in men and women. Results from the literature dealing with this issue and the implications for the present study should be clearly discussed.

2. Social context

For the cooking habit, further details of the social context have to be indicated. I can take two examples before smell loss. In case of a single mother with her children, cooking is mandatory, whatever the pleasantness. On the contrary, a single man can use delivery, take off, and cook rarely.

3. Trigeminal sensitivity

Authors should further discuss trigeminal chemesthesis by using mouthfeel score (figure 3) but also consumption of condiments and spices (figure 4). Is there a compensation in case of olfactory distortion or loss?

4. BMI

Olfactory perception was shown to be decreased with increased BMI. This issue should be addressed in the questionnaire and the MS. In that sense, how was body weight estimated? For example, what is the frequency of weighing? This is a crucial issue as well as the estimation of severity in parosmia (see 5).

5. Advantages and limits of self-assessment

There should be a paragraph about this issue in the discussion. In the absence of clinical testing using standard techniques such as the Sniffin’ Stick Test to probe the severity of parosmia and anosmia, conclusions should be drawn extremely carefully. “3.5.2. Parosmia severity”: authors should be careful. Estimation of distortion by the frequency (rare to always) does not mean that odor confusion is elevated between odors and which categories of odors are implicated (the quality of parosmia). Let’s imagine that olfactory distortion does not concern (so much) food odors, then the overall conclusion would be weakened.

Author Response

Dear reviewer,

Thank you for taking your time to review our article and for giving us feedback to improve our paper for submission. We have read and reviewed your comment and advice, which will be addressed below.  

Kind regards,

The corresponding author

  1. Gender

Table 1 clearly shows that 80% of the 522 subjects are women, thus the results of this study present a massive bias towards female odor perception. Surprisingly, this fact is not discussed at any time in the MS. It has to be, since olfactory threshold, discrimination and identification are different in men and women. Results from the literature dealing with this issue and the implications for the present study should be clearly discussed.

Reply: This is a good point. Analysis on gender differences has been added to the results section and a paragraph has been added to address this point in the discussion.

  1. Social context

For the cooking habit, further details of the social context have to be indicated. I can take two examples before smell loss. In case of a single mother with her children, cooking is mandatory, whatever the pleasantness. On the contrary, a single man can use delivery, take off, and cook rarely. 

Reply: We agree that several factors influence can influence cooking habits. Due to the length of the questionnaire and all of the potential influencing factors, not all of these potential factors were included in our data collection and analysis. We focused on having an age-matched control group with female overweight to ensure that the core demographics where comparable. We have included this point on social context in the limitations section.

  1. Trigeminal sensitivity

Authors should further discuss trigeminal chemesthesis by using mouthfeel score (figure 3) but also consumption of condiments and spices (figure 4). Is there a compensation in case of olfactory distortion or loss?

Reply: Interesting question, thanks! We have added a further analysis in the results (3.6.2 Liking). However, no differences in liking were significantly associated with changes in rated mouthfeel awareness.

  1. BMI

Olfactory perception was shown to be decreased with increased BMI. This issue should be addressed in the questionnaire and the MS. In that sense, how was body weight estimated? For example, what is the frequency of weighing? This is a crucial issue as well as the estimation of severity in parosmia (see 5).

Reply: We have only included self-reported weight change due to olfactory dysfunction in the manuscript, not BMI. This has been clarified in the methods section and limitations.  

  1. Advantages and limits of self-assessment

There should be a paragraph about this issue in the discussion. In the absence of clinical testing using standard techniques such as the Sniffin’ Stick Test to probe the severity of parosmia and anosmia, conclusions should be drawn extremely carefully. “3.5.2. Parosmia severity”: authors should be careful. Estimation of distortion by the frequency (rare to always) does not mean that odor confusion is elevated between odors and which categories of odors are implicated (the quality of parosmia). Let’s imagine that olfactory distortion does not concern (so much) food odors, then the overall conclusion would be weakened.

Reply: We agree that this is an important aspect to include. This has been added to the limitations section.

Reviewer 2 Report

The manuscript gives a useful knowledge to the physicians treating olfactory impaired patients. I wish if the comments would help to improve the manuscript.

1.     The major concerns are the heterogeneous etiologies of olfactory dysfunction as discussed. It would be better to show the analysis to compare only patients after COVID-19 infection and normal subjects.

2.     3-D figures are difficult to interpret a little, although they look nice.

Author Response

Dear reviewer,

Thank you for taking your time to review our article and for giving us feedback to improve our paper for submission. We have read and reviewed your comment and advice, which will be addressed below.  

Kind regards,

The corresponding author

  1. The major concerns are the heterogeneous etiologies of olfactory dysfunction as discussed. It would be better to show the analysis to compare only patients after COVID-19 infection and normal subjects.

Reply: We agree that the heterogeneous etiologies does have some disadvantages, however, in earlier investigations on parosmic patients, post-COVID and pre-COVID parosmia were found to be similar in TDI-scores and gas chromatography-olfactometry scores [Parker et al, ‘Molecular Mechanism of Parosmia’, 2021]. To address the raised concern and to test if parosmia differs between post-COVID and non-COVID individuals, we added further analysis on effects of COVID within the parosmic group. We think this is a valuable addition to the manuscript, and thank the reviewer for this comment.

  1. 3-D figures are difficult to interpret a little, although they look nice.

Reply: We have updated the figure legends of figure 2 to ease readability.